# Viruses Infecting Greenhood Orchids (Pterostylidinae) in Eastern Australia

**DOI:** 10.3390/v14020365

**Published:** 2022-02-10

**Authors:** Hsu-Yao Chao, Mark A. Clements, Anne M. Mackenzie, Ralf G. Dietzgen, John E. Thomas, Andrew D. W. Geering

**Affiliations:** 1Centre for Horticultural Science, Queensland Alliance for Agriculture and Food Innovation, The University of Queensland, St. Lucia, QLD 4072, Australia; r.dietzgen@uq.edu.au (R.G.D.); j.thomas2@uq.edu.au (J.E.T.); 2Centre for Australian National Biodiversity Research, National Facilities and Collections, Commonwealth Scientific and Industrial Research Organisation, Canberra, ACT 2601, Australia; mark.clements@csiro.au; 3Independent Researcher, Hawker, ACT 2614, Australia; annem5349@gmail.com

**Keywords:** native vegetation, disease, surveillance, virus discovery, cryptic virus, mycovirus, ssRNA virus, dsRNA virus, *Tospoviridae*, *Solemoviridae*, thrips

## Abstract

The Australasian biogeographic realm is a major centre of diversity for orchids, with every subfamily of the Orchidaceae represented and high levels of endemism at the species rank. It is hypothesised that there is a commensurate diversity of viruses infecting this group of plants. In this study, we have utilised high-throughput sequencing to survey for viruses infecting greenhood orchids (Pterostylidinae) in New South Wales and the Australian Capital Territory. The main aim of this study was to characterise Pterostylis blotch virus (PtBV), a previously reported but uncharacterised virus that had been tentatively classified in the genus *Orthotospovirus*. This classification was confirmed by genome sequencing, and phylogenetic analyses suggested that PtBV is representative of a new species that is possibly indigenous to Australia as it does not belong to either the American or Eurasian clades of orthotospoviruses. Apart from PtBV, putative new viruses in the genera *Alphaendornavirus*, *Amalgavirus*, *Polerovirus* and *Totivirus* were discovered, and complete genome sequences were obtained for each virus. It is concluded that the polerovirus is likely an example of an introduced virus infecting a native plant species in its natural habitat, as this virus is probably vectored by an aphid, and Australia has a depauperate native aphid fauna that does not include any species that are host-adapted to orchids.

## 1. Introduction

Research on plant viruses has mainly focused on the ones that cause diseases in food and fibre crops, while those that infect plants within natural ecosystems are generally poorly understood unless located at the interface with agriculture [1,2]. Plant viruses likely play significant ecological roles, as plant pathogens in general are important determinants of plant community structure, as well as individual plant species diversity [3]. Even a slight change in the fitness of a plant species could affect the abundance of its genotype over longer timeframes. While plant viruses are conventionally regarded as pathogens and therefore detrimental to the host plant, some viruses may actually confer fitness benefits such as drought tolerance [4].

The Orchidaceae is the second largest family of plants in the world and constitutes about 8% of all currently accepted plant species [5]. Biogeographical studies suggest that orchids originated on the Australian part of Gondwana about 112 million years ago, after the split of Africa and India/Madagascar, then migrated to South America via the Antarctic land bridge by about 90 million years ago [6]. More recent migrations are thought to have occurred following the collision of tectonic plates, such as that of the Australian and Sunda plates, and by infrequent long-distance dispersal of the dust-sized seed in jet streams across oceans. The Orchidaceae is divided into five subfamilies, and then into tribes and subtribes [7]. Representatives of all subfamilies naturally occur in the Australasian biogeographic realm, encompassing Australia, New Guinea and the eastern part of the Indonesian archipelago [6].

Tribe Cranichideae (subfamily Orchidoideae) is a cosmopolitan group of mainly terrestrial species comprising nine subtribes in two main areas of distribution. There are six subtribes represented in South America while members of four subtribes inhabit the Australasian region: the Pterostylidinae, 12 genera from the Goodyerinae, a few species from the Spiranthinae and the endemic Achlydosinae from New Caledonia [8]. There is also one endemic subtribe in West Africa. Subtribe Pterostylidinae is predominantly found in Australia and is disproportionately species-rich, containing 312 named species, which amounts to 14% of all taxa in the Cranichidieae [9,10]. The Pterostylidinae is further divided into 10 genera (*Speculantha*, *Diplodium*, *Pterostylis*, *Bunochilus*, *Hymenochilus*, *Oligochaetochilus*, *Pharochilum*, *Plumatichilos*, *Stamnorchis* and *Urochilus*), the majority of which occur in southern Australia, where their main growing season is during winter and early spring [8,10]. During the hot dry summer, these orchids die back to underground tubers, from which plants regenerate when the rains return in late autumn. The Pterostylidinae have special symbiotic relationships with mycorrhizal fungi such as *Ceratobasidium* spp., which are vital for successful seed germination [11,12]. The flowers of these orchids emit sex pheromones that attract male fungus gnats, which attempt to copulate with the flower and in so doing, inadvertently transfer pollen between plants [13].

Thus far, surveys from Australia suggest that the types of viruses infecting orchids in cultivation are very different to those growing in the wild. Odontoglossum ringspot virus (genus *Tobamovirus*) and Cymbidium mosaic virus (genus *Potexvirus*) are the most common viruses of cultivated orchids, and these are mechanically transmissible, highly infectious and occur in most countries where orchids are grown [14,15,16]. Orchid fleck virus (genus *Dichorhavirus*), another cosmopolitan pathogen with a false spider mite (*Brevipalpus* spp.) vector [17], is also frequently detected in cultivated orchids in Australia [16]. In contrast, the viruses infecting orchids in natural environments appear to be mostly indigenous, although there are instances of infections of terrestrial orchids in temperate south-west Western Australia by introduced viruses such as bean yellow mosaic virus (genus *Potyvirus*), Ornithogalum mosaic virus (genus *Potyvirus*) and turnip yellows virus (genus *Polerovirus*) [18]. Examples of orchid-infecting viruses that are likely indigenous to Australia include Diuris virus Y (DiVY), blue squill virus A, donkey orchid virus A, Diuris virus A, Diuris virus B, Diuris pendunculata cryptic virus, donkey orchid symptomless virus and Caladenia virus A [18,19,20]. Mycoviruses have also been found in the mycorrhizal fungi associated with orchids [21].

One interesting discovery of a virus in a natural population of orchids in Australia has been Pterostylis blotch virus (PtBV) [15]. PtBV was tentatively classified in the genus *Orthotospovirus* (family *Tospoviridae*, order *Bunyavirales*) based on virion morphology and amplification of cDNA using a universal tospovirus reverse transcription (RT)-PCR assay targeting the RNA polymerase gene on genome segment L. Unfortunately, no sequence information for this virus was deposited in the international nucleotide sequence databases. PtBV was found at five different locations in the Australian Capital Territory (ACT), inland New South Wales (NSW) and Victoria, and in six orchid species, *Pterostylis curta* R.Br., *Pterostylis hispidula* Fitzg., *Plumatichilos plumosa* Szlach., *Diplodium reflexum* (R.Br.) D.L.Jones & M.A.Clem., *Diplodium revolutum* (R.Br.) D.L.Jones & M.A.Clem., and *Speculantha parviflora* (R.Br.) D.L.Jones & M.A.Clem. Orthotospoviruses are renowned for having extremely wide host ranges, numbering over 1000 species in 15 monocotyledenous plant families, 69 dicotyledenous plant families and one family of pteridophytes in the case of the tomato spotted wilt virus (TSWV) [22]. Hence, PtBV may represent an extension in the host range of an uncharacterised orthotospovirus in ornamental or crop plant species. Three orthotospoviruses are already widely distributed in Australia, namely TSWV, capsicum chlorosis virus (CaCV) and iris yellow spot virus (IYSV), and all are regarded as introduced pathogens [23]. The two centres of diversity of orthotospoviruses are the Americas and Eurasia [24].

In this work, we have undertaken surveys of *Pterostylis nutans* R.Br., *P. curta* and *Oligochaetochilus hamatus* (Blackmore & Clemesha) Szlach. in eastern Australia to identify and characterise the viruses infecting these native plant species using high-throughput sequencing methods. Priority was given to characterise PtBV and therefore localities where this virus had been historically recorded were revisited. However, during these surveys, alphaendorna-, amalga-, polero-, totiviruses were serendipitously discovered, and many of these were records of novel viruses.

## 2. Materials and Methods

### 2.1. Surveys and Plant Samples

Entire plants of *P. nutans*, *P. curta* and *O. hamatus* were collected from four sites within the ACT and seven sites within NSW (Table 1). Leaf samples were either processed immediately upon delivery of the plants to the laboratory at the Ecosciences Precinct, Dutton Park, Queensland or the plants were transplanted into pots containing a sand-peat mix and grown in an insect-free glasshouse at the same location for later sampling. Permission to collect the plants was provided by NSW Scientific Licence SL100750 and ACT Licence No. PL2017136.

Total RNA was extracted from fresh or lyophilised leaves using either a TRIzol™ Plus RNA Purification Kit (Thermo Fisher Scientific, Waltham, MA, USA) or an ISOLATE II RNA Plant Kit (Meridian Bioscience, Cincinnati, OH, USA) using the RLY buffer as per the manufacturer’s instructions.

### 2.2. High-Throughput Sequencing and Sequence Assembly

Total RNA extracts from each orchid plant were submitted to the Australian Genome Research Facility (Melbourne, Australia) and RNA quality checks, library preparation and high-throughput sequencing (HTS) were carried out there. Paired-end cDNA libraries were prepared from the RNA samples using a TruSeq^®^ Stranded Total RNA Library Preparation Kit with Ribo-Zero™ Plant (Illumina, San Diego CA, USA). Sample 13365 was sequenced using a NovaSeq 6000 System with NovaSeq 6000 SP Reagent Kit v1 (200 cycles, Illumina) to generate 100 bp pair-end reads, and data from this sample were obtained from three lanes of the flow cell. The samples GF and HR were sequenced using the MiSeq System with a MiSeq Reagent Kits v2 (300 cycles, Illumina) to generate 150 bp pair-end reads. The other samples were sequenced using the NovaSeq 6000 System with NovaSeq 6000 SP Reagent Kit v1.5 (300 cycles, Illumina) to generate 150 bp pair-end reads.

The raw reads generated from HTS were trimmed with Trimmomatic (version 0.39) [25]. The sequence file used for adapter clipping was TruSeq3-PE-2.fa. The maximum number of seed mismatches for adapter clipping was set to 2, palindrome clip threshold to 30, simple clip threshold to 7 and minimum length of detected adapters to 1. Quality trimming was carried out by setting the quality thresholds for trimming leading bases, trimming trailing bases and sliding window (window size = 4) to 20. For *de novo* assembly, the minimum read length was set to 100 bp or 130 bp; for read mapping, the minimum read length was set to 30 bp.

*De novo* sequence assembly was executed with SPAdes v3.11.1 [26] using the parameters listed in Appendix A, with “read error correction” and “mismatch correction” enabled. Assembled contigs were queried against a local protein database built from NCBI complete RefSeq release (Release 201) of viral sequences using BLASTX (version 2.3.0+) [27]. Plant-associated virus sequence contigs were then manually identified based on significant alignment results.

Paired reads at least 30 bp in length after trimming were mapped against the near complete virus genome sequences using BWA (version 0.7.13) with backtrack algorithm [28] and default settings. Variants were called on the mapping output using SAMtools (version 1.10) [29] and BCFtools (version 1.3) [29] to obtain consensus sequences and the coverage of each position in the reference sequences. The variant caller was consensus caller (bcftools call -c option), and the *p*-value threshold for accepting a variant (bcftools call -p option) was set to 0.05. Domains of viral polyproteins were predicted by querying the open reading frame against the NCBI Conserved Domain Database (version 3.19) using the search function with default settings [30,31]. Putative ribosomal frameshift sites were manually identified by searching for previously identified slippery sequences at feasible positions in the newly discovered virus genomes. The annotations and organisations of verified virus genomes were then visualised in Geneious Prime version 2020.2.3 (https://www.geneious.com, last accessed on 16 January 2022).

To confirm the presence of each virus discovered by HTS, RT-PCR assays were done using a OneTaq^®^ One-Step RT-PCR Kit (New England Biolabs, Ipswich MA, USA) as per the manufacturer’s instructions and with the following thermocycling parameters: 15 min at 48 °C, 1 min at 94 °C followed by 40 cycles of 15 s at 94 °C, 30 s at 60 °C and 1 min at 68 °C, ending with 5 min of final extension at 68 °C. The primers used for RT-PCR and amplicon sequencing are described in Appendix A. The amplicons were gel-purified and directly sequenced by Macrogen (Seoul, South Korea).

### 2.3. Rapid Amplification of cDNA Ends (RACE)

For the single-stranded RNA viruses, 5′ and 3′ RACE were carried out using *c.* 1 μg of a DNase I-treated total plant RNA extract as the starting material. All gene-specific primers used in the PCR and sequencing are listed in the Appendix A). The RNA was polyadenylated using *E. coli* poly(A) polymerase (New England Biolabs) and cleaned up with a Monarch^®^ RNA Cleanup Kit (10 µg, New England Biolabs). cDNA was synthesised from the polyadenylated RNA using Maxima H Minus Reverse Transcriptase (Thermo Fisher Scientific) as per the manufacturer’s instructions, except that the custom primer 5′-CCACGCGTATCGATGTCGAC(dT)40VN-3′ was used instead of the recommended oligo(dT)18 primer. The synthesised cDNA was purified using a Monarch^®^ PCR & DNA Clean-up Kit (5 μg, New England Biolabs) following the protocol for ssDNA clean up. The cDNA was then polyadenylated using Terminal Transferase (New England Biolabs), assuming the ratio of 3′ cDNA ends to dATP was 1: >5000. For 3′ RACE, the first round of PCR was done with a gene-specific forward primer and the reverse primer 5′-CCACGCGTATCGATGTCGAC-3′. The PCR amplicon was diluted and used as the template for nested PCR using a new gene-specific forward primer and the same reverse primer. The target amplicon was gel-purified and sequenced with another nested gene-specific forward primer. For 5′ RACE, the first round of PCR was completed using the forward primer 5′-GGCCACGCGTCGACTAGTAC(dT)40VN-3′ and a gene-specific reverse primer. The PCR amplicon was diluted and used as the template for nested PCR, using 5′-GGCCACGCGTCGACTAGTAC-3′ as the forward primer and a nested gene-specific reverse primer. The target amplicon was gel-purified and sequenced with another nested gene-specific reverse primer. Platinum™ SuperFi II Green PCR Master Mix (Thermo Fisher Scientific) was used for all PCRs and thermocycling conditions were 30 s at 98 °C followed by 35 cycles of 10 s at 98 °C, 10 s at 60 °C and 30 s at 72 °C, ending with 5 min of final extension at 72 °C.

For samples containing dsRNA viruses or viruses with a stable replicative dsRNA form, the ends of the virus genomes were determined using 5′ RACE in a similar manner as described above, except that cDNA was synthesised from about 150 ng of total plant RNA using gene-specific primers (Appendix A), and all subsequent PCRs were carried out using Q5^®^ High-Fidelity 2× Master Mix (New England Biolabs) with the following thermocycling conditions: 30 s at 98 °C followed by 40 cycles of 5 s at 98 °C, 10 s at 55 °C or 60 °C and 10 s at 72 °C, ending with 2 min of final extension at 72 °C. The gel purification and direct sequencing of target amplicons were mostly outsourced to Macrogen (Seoul, South Korea), with some samples gel-purified in-house using Monarch^®^ DNA Gel Extraction Kit (New England Biolabs) and sequenced by the Australian Genome Research Facility (Melbourne, Australia).

### 2.4. Phylogenetic Analyses and Pairwise Sequence Comparisons

Predicted amino acid sequences were aligned using COBALT [32], as provided by NCBI (https://www.ncbi.nlm.nih.gov/tools/cobalt/, last accessed on 16 January 2022). The gap opening penalty, gap extension penalty, end-gap opening penalty and end-gap extension penalty were set to −6, −6, −1 and −1, respectively. The accession numbers of the sequences included in the phylogenetic analyses are listed in the Appendix A). After manually excluding non-conserved or poorly aligned regions at the N- and C-termini, the aligned protein sequences were back-translated to codon sequence alignments for the phylogenetic analyses. FASconCAT-G [33] was used to concatenate the sequence alignments, and maximum likelihood (ML) trees were reconstructed using IQ-TREE (version 1.6.12 or 2.1.3) [34]. The nucleotide substitution models were selected based on corrected Akaike information criterion or Bayesian information criterion according to the model test results obtained using ModelFinder [35]. The option of more thorough and slower nearest neighbour interchange search (-allnni), was enabled for tree search. If the chosen substitution model included + I + G (invariable site plus discrete gamma model), the option -opt-gamma-inv, was enabled for more thorough estimation of + I + G model parameters. Branch support was assessed by non-parametric bootstrapping, using 1000 samples from the original alignment. The starting seed number for all analyses was arbitrarily specified as 109,843. All other parameters used by IQ-TREE remained by default. For phylogenetic analyses performed with a concatenated sequence alignment, gene trees were inferred, and gene concordance factors (gCF) were calculated as per the IQ-TREE manual. The gCF were then manually converted to the number of decisive gene trees that support the species tree. The nodes with a bootstrap value of less than 50 in the ML trees were collapsed using TreeGraph (version 2.15.0-887 beta) [36] and visualised with FigTree (version 1.4.4, http://tree.bio.ed.ac.uk/software/figtree/, last accessed on 16 January 2022).

For calculating pairwise sequence identities, nucleotide (nt) sequences were aligned with Clustal Omega 1.2.2 [37] using Geneious Prime version 2020.2.3 with automatic parameter adjustment enabled, while amino acid (aa) sequences were aligned using COBALT [32] as described in the previous paragraph.

## 3. Results

Virus-like sequence contigs were identified in 12 of the 15 orchid plants that were analysed by HTS: samples 13402, PI-1 and PI-2 were apparently uninfected (Table 1). All tested plants grew in undisturbed natural habitats, except for the two plants from Pine Island (PI-1 and PI-2), which were found at a popular recreational reserve on the banks of the Murrumbidgee River (Table 1). The numbers of raw reads generated from HTS for each sample and the numbers of reads left after trimming are shown in Appendix A. The viruses identified in the greenhood orchids in this study had single-stranded, positive-sense (+) or negative-sense/ambisense (−) RNA, or double-stranded (ds) RNA genomes. They could be taxonomically classified into five genera in five different families: (+) RNA: *Alphaendornavirus* (family *Endornaviridae*), *Polerovirus* (family *Solemoviridae*); (−) RNA: *Orthotospovirus* (family *Tospoviridae*); dsRNA: *Amalgavirus* (family *Amalgaviridae*), *Totivirus* (family *Totiviridae*), as detailed below.

The presence of the virus identified by HTS in each of the plant samples was confirmed by RT-PCR. The sequences of each of the amplicons matched the corresponding sequences obtained by HTS with 100% nt sequence identity.

### 3.1. Molecular Characterisation of PtBV

PtBV was first reported in 2000, infecting a colony of *P. nutans* growing in dry sclerophyll forest on the slopes of Black Mountain in the ACT (MA Clements and AJ Gibbs, pers. comm.). This colony of orchids has persisted and when inspected in 2020, chlorotic blotch symptoms were observed on some but not all of the orchids (Figure 1). A symptomatic plant was collected from this location (sample 13365) and chosen for the first round of HTS.

After assembling the HTS data, three contigs were obtained that contained complete coding sequences for proteins that are homologous to those of orthotospoviruses based on BLASTX searches of the NCBI RefSeq protein database. These virus sequences were assumed to be representative of PtBV, hence the name was retained but since the sequence of the PCR product obtained by Gibbs and colleagues [15] was not deposited in one of the international nucleotide sequence databases, sequence comparisons were not possible to link these virus discoveries. Following 5′ and 3′ RACE, the complete genome segment sequences are 8636 nt for the L segment, 4697 nt for the M segment and 3159 nt for the S segment and these have been deposited in GenBank with accession numbers OL471332 to OL471334, respectively. The genome organisation resembled by the complete sequences of the assumed PtBV from the orchid sample 13365 (PtBV-13365) is typical of an orthotospovirus (Figure 2): one negative-sense long open reading frame (ORF) in the L RNA segment encoding the putative RNA-dependent RNA polymerase (RdRp); one positive-sense ORF encoding the putative non-structural movement protein (NSm) and one negative-sense ORF encoding the putative glycoprotein precursor (GP) at the 5′ and 3′ ends of the M RNA segment, respectively, and one positive-sense ORF encoding the putative non-structural RNA silencing suppressor (NSs) and one negative-sense ORF encoding the putative nucleocapsid protein (N) at the 5′ and 3′ ends of the S RNA segment, respectively.

The RdRp and N protein coding regions, at 8529 and 735 nt, respectively, are shorter than those of any other known orthotospoviruses. Conserved and complementary sequence motifs were present at the termini of the three genome segments: 5′-AGAGC-3′ at the 5′ end and 5′-GCUCU-3′ at the 3′ end. PtBV was also detected in *P. nutans* and *P. curta* at two other sites in the ACT (samples GF and HR, Table 1), and in *P. curta* from Warrumbungle National Park (sample 13399, Table 1), which is located approximately 550 km north of the ACT. These additional three infected plants also showed symptoms similar to those shown in Figure 1, but such symptoms were not observed in other samples where PtBV was not detected.

Pairwise sequence comparisons showed that all the recovered genomes of PtBV have less than 40% aa sequence identity in the N protein to any other recognised or tentative orthotospovirus species (Appendix A) but greater than 99% aa sequence identity to each other (Appendix A), suggesting that PtBV should be classified in a novel species according to the International Committee on Taxonomy of Viruses (ICTV) species demarcation threshold of 90% aa sequence identity [38]. PtBV-13399 was the most distantly related to the designated exemplar of PtBV from Black Mountain (PtBV-13365, Appendix A), likely reflecting the fact that it was the most geographically separated among the PtBV that were detected. The genome architectures of all PtBV were identical, except that the lengths of the intergenic regions (IGR) in the M and S RNA segments varied. The lengths of the IGR in the M segments of PtBV-13365, -GF, -HR and -13399 are, respectively, 242, 240, 194 and 253 nt, while those in the S segments are, respectively, 969, 1093, 1100 and 841 nt. The genome segments of PtBV-13399, -GF and -HR, excluding the 5′ and 3′ untranslated regions (UTRs), were deposited in GenBank with accession numbers OL471335 to OL471343.

In a phylogenetic reconstruction based on the *RdRp* codon sequence alignment, PtBV fell within a monophyletic clade containing all recognised orthotospoviruses, within which groundnut chlorotic fan-spot virus (GCFSV) was the basal taxon (Appendix A). To provide a genome-wide estimation of phylogeny, a second analysis was carried out using a concatenated codon sequence alignment from five of the orthotospovirus genes (*RdRp*, *NSm*, *GP*, *NSs* and *N*). PtBV and Barleria chlorosis-associated virus (BCaV) were shown to be sister taxa with 100% bootstrap support for the branch node and genealogical concordance with all but the NSs gene (Figure 3 and Appendix A). Considering that the currently recognised clades of orthotospoviruses described in the literature, i.e., the American and the Eurasian clades, are well supported by the species tree with 100% bootstrap support and 100% genealogical concordance (all five genes, Figure 3), BCaV and PtBV should probably be treated as distinct lineages given that they do not have the same level of genealogical concordance and that they share low sequence similarities.

### 3.2. Pterostylis Alphaendornavirus

A long sequence contig from *P. nutans* sample HR had a significant match to winged bean endornavirus 1 (WBEV1, genus *Alphaendornavirus*, family *Endornaviridae*) following a BLASTX search of the NCBI RefSeq protein database. This virus, which was provisionally named Pterostylis alphaendornavirus (PtAEV), was not found in any other orchid sample. Members of *Endornaviridae* are known to have a stable dsRNA replicative form. Utilising this feature and following 5′ RACE, a complete genome sequence was obtained (GenBank accession number OL471320), which consists of 14,889 nt and has a single, long ORF of 14,835 nt that is flanked by a 5′ UTR of 22 nt and a 3′ UTR of 32 nt (Figure 2). As is typical of members of the family *Endornaviridae*, there is a 10-base stretch of cytosines at the 3′ end of the PtAEV genome. Following conceptual translation of the ORF, a 560 kDa protein was predicted that contains four conserved domains, which in order from the N- to the C-terminus, are cysteine proteinase (CD accession number pfam05412), helicase (CD accession number COG1112), glycotransferase (CD accession number cd03784) and RdRp (CD accession number pfam00978). A BLASTP search of the NCBI non-redundant protein sequence database revealed WBEV1 (GenBank accession number YP_009305414.1) as the closest match, with 72% query coverage and 30.8% pairwise sequence identity.

In a phylogenetic reconstruction using the codon sequence alignment from the *RdRp*, PtAEV was sister to a clade of plant-infecting alphaendornaviruses containing WBEV1, Phaseolus vulgaris endornavirus 2 (PvEV2), bell pepper endornavirus (BPEV) and hot pepper endornavirus (HPEV, Figure 4). Over the entire length of the genome, PtAEV has less than 40% nt sequence identity to these viruses, which is well below the 75% nt sequence identity demarcation threshold for distinguishing different species in this genus [39].

### 3.3. Pterostylis Amalgaviruses

Amalgaviruses (genus *Amalgavirus*, family *Amalgaviridae*) have linear dsRNA genomes of about 3.5 kb with two overlapping ORFs. Amalgavirus sequence contigs were detected in nine *Pterostylis* plants (Table 1) and two sequence clusters were identified based on an 85% nt sequence identity cut-off (Appendix A). One cluster was represented by a single sequence, PtAV-13438-2, and the other contained the remaining 10 sequences. PtAV-13365 and PtAV-13438-2 were selected as exemplars for two tentative new amalgaviruses that were, respectively, named Pterostylis amalgavirus 1 (PtAV1) and Pterostylis amalgavirus 2 (PtAV2). After 5′ RACE was carried out, complete genome sequences of these viruses were obtained, which comprised 3428 nt for PtAV-13365 and 3426 nt for PtAV-13438-2 (GenBank accession numbers OL471321 and OL471331, respectively). Both amalgavirus genomes contain two overlapping ORFs that are predicted to produce a fusion protein through a + 1 ribosomal frameshift (Figure 2). The same slippery sequence motif, UUUCGU, where the predicted ribosomal frameshift occurs at the C, was identified at the 3′ end of ORF1 in both amalgavirus genomes. The ORF2 gene product, likely expressed through the + 1 ribosomal frameshift in both genomes, has an RdRp domain, suggesting it functions as a polymerase. The ORF1 gene product is predicted to be the capsid protein [40], although no actual virions of amalgaviruses have been observed to date.

In a phylogenetic reconstruction using the *RdRp* sequence alignment, both PtAV1 and PtAV2 clustered within the genus *Amalgavirus* but in different positions of the tree (Figure 5). PtAV1 formed its own distinctive clade, while PtAV2 clustered with southern tomato virus (STV), Allium cepa amalgavirus 1 (AcAV1), Allium cepa amalgavirus 2 (AcAV2), rhododendron virus A (RhVA), spinach amalgavirus 1 (SpAV1) and Vicia cryptic virus M (VCVM, Figure 5). Species demarcation criteria have yet to be set for the genus *Amalgavirus*. However, members of the family *Partitiviridae* have a homologous RdRp [40], and the species demarcation criterion for species in this family is ≤90% aa sequence identity in the RdRp [41]. As PtAV1 and PtAV2 have not more than 60% aa identity to each other and to any other known amalgavirus species in the *RdRp* (Appendix A), we suggest that they each should be classified in a new species in the genus. The genomes of the other nine PtAV1, each 3175 nt in length excluding the 5′ and 3′ UTRs, were deposited in GenBank with accession numbers OL471322 to OL471330.

### 3.4. Pterostylis Totiviruses

Five totivirus sequence contigs, hereafter referred to as Pterostylis totivirus (PtTV), were detected in three *P. nutans* (samples HR, 13365, 13421) and one *P. curta* (sample GF) plants (Table 1). All PtTV genomes contain two non-overlapping ORFs that are predicted to produce a fusion protein through a − 1 ribosomal frameshift, enabled by a slippery sequence motif (GGAUUUU) at the 3′ end of ORF 1, where the rightmost U is the predicted ribosomal frameshift site (Figure 2). The ORF1 gene product is predicted to be the coat protein, while the fusion protein is predicted to be the RdRp (Figure 2). PtTV-13421-2 has less than 65% nt sequence identity to the other totivirus sequence contigs in the coding region, while the other sequences were more than 80% identical to each other (Appendix A). Following 5′ RACE, the complete genome sequences of PtTV-13365 and PtTV-13421-2 were 5003 and 4972 nt, respectively, and these have been deposited in GenBank with accession numbers OL471345 and OL471347, respectively. The remaining three sequence contigs for PtTV-HR, PtTV-13421-1 and PtTV-GF, each 4847 nt long after removing the unverified 5′ and 3′ UTRs, represented near-complete genomes (GenBank accession numbers OL471349, OL471346 and OL471348, respectively). Although the 5′ and 3′ UTRs of PtTV-13365 and PtTV-13421-2 are different in length, the arrangement of the coding regions are identical (Figure 2).

Different totiviruses are defined as having less than 50% aa sequence identity in the fusion protein, although this is not a canonical rule [42]. All five *Pterostylis*-associated totivirus ORF2 sequences have nearly 70% aa sequence identity to a totivirus-like sequence identified in *Pterostylis* from Western Australia, named Pterostylis sanguinea totivirus A (PtSTVA, Appendix A), suggesting that they could be classified as the same virus species even though the current sequence data of PtSTVA are incomplete. Based on a phylogenetic analysis using the *RdRp* codon sequence, PtSTVA and PtTV formed a distinct clade within the genus *Totivirus*, which is most closely related to Saccharomyces cerevisiae virus L-A (ScVLA) and Tuber aestivum virus 1 (TaV1, Figure 6).

### 3.5. Pterostylis Polerovirus

A 5757-nt sequence (GenBank accession number OL471344) of a polerovirus, tentatively named Pterostylis polerovirus (PtPV), was discovered in sample 13365 (Table 1). The genome organization of PtPV is typical of poleroviruses in general (Figure 2), with seven predicted ORFs altogether translated into seven protein products, whose putative functions are as follows: RNA silencing suppressor (P0); serine peptidase (P1); RNA-dependent RNA polymerase (P1–P2 fusion protein); major capsid protein (P3); movement protein (P4); minor capsid protein (P3–P5 read-through protein); and accessory movement protein (P3a) [43]. P0 and P1 are encoded by ORF0 and ORF1, respectively. A slippery sequence motif, GGGAAAC, is present in ORF1, where C is the predicted ribosomal frameshift site; by utilization of this motif, a − 1 ribosomal frameshift would occur, resulting in the co-translation of ORF2 into the P1–P2 RdRp fusion protein. ORF3a encoding P3a starts with a non-canonical codon, AUU. ORF4 encoding P4 is nested within ORF3, which encodes P3. ORF5 is co-translated with ORF3 through stop-codon readthrough into P3–P5 read-through protein.

A reconstruction of phylogeny based on the concatenated alignments of four conserved regions (genes) in the protein coding sequences, one in ORF1, one in ORF2, one in ORF3 and the other encoding the protein sequence at the C-terminus of P3 as part of the P3–P5 read-through protein, grouped PtPV with potato leafroll virus (PLRV), cereal yellow dwarf virus-RPS (CYDV-RPS) and cereal yellow dwarf virus-RPV (CYDV-RPV) but this relationship was only supported by three out of the four gene trees (Figure 7). Contrary to PtPV and the CYDVs, PLRV appears to have a different origin in the fourth conserved region (Appendix A). On the other hand, the node accommodating PtPV and the CYDVs in the species tree was not supported by any gene tree (Figure 7), reflecting that there is no node in any of the four gene trees containing “only” PtPV and the CYDVs (Appendix A).

The predicted protein products of PtPV all have less than 70% aa sequence identity to their respective closest relatives (Appendix A), suggesting that PtPV should be classified in a new species in the genus *Polerovirus* given the species demarcation criterion of more than 10% difference in the aa sequence identity of any gene product [44].

## 4. Discussion

Viruses belonging to five different genera were identified during this survey of greenhood orchids, although only PtBV and PtPV, as members of the genera *Orthotospovirus* and *Polerovirus*, respectively, are likely to be pathogenic. Furthermore, PtBV is probably the main cause of the yellow blotch symptoms, as it was the only virus consistently associated with these distinctive symptoms (Figure 1). The majority of alphaendornaviruses cause latent infections, although some may produce mild symptoms, such as balloon flower endornavirus, whose presence correlates with leaf shrinkage in *Platycodon grandiflorus* [45]. Amalgaviruses are cryptic in nature, occur at low titres and are only thought to be able to propagate within their host plant by cell division as they lack a movement protein gene [46].

Orchids have well-known symbiotic relationships with both mycorrhizal and non-mycorrhizal fungi, which assist with nutrient uptake, modulate plant hormone levels and confer defence against fungal pathogens [47]. Using our HTS strategy, it is not possible to definitively conclude whether some of the viruses that were discovered were infecting the plant or a fungal species that might be associated with the plant. Similar to the totivirus detected in this study, Ong, et al. [48] detected Pterostylis sanguinea totivirus A in leaf tissues of *Pterostylis* plants but given that all well-characterised totiviruses have fungal hosts (Virus Taxonomy: 2020 Release, https://talk.ictvonline.org/taxonomy/, last accessed on 16 January 2022), it would seem most likely this totivirus was infecting an endophytic fungus within the plant. The various known alphaendornaviruses may have plant, fungal or oomycete hosts but given that PtAEV was most closely related to the plant-infecting species, it is likely that this virus also shares the same ecological niche.

The recognition of PtBV as belonging to a distinct species within the genus *Orthotospovirus* is strongly supported by results of genome organization, phylogenetic relationships and N protein sequence comparisons. PtBV is only the fifth virus from this taxon to have been found in Australia after TSWV, CaCV, IYSV and impatiens necrotic spot virus [23,49]. These four orthotospoviruses are broadly distributed around the world and are almost certainly recent arrivals [23], whereas PtBV may represent the first indigenous orthotospovirus from Australia. Supporting this conclusion, PtBV represents a separate distinct lineage in the orthotospovirus phylogenetic tree. Secondly, the virus was found in undisturbed habitats that were protected within national parks, far removed from any agricultural areas. Furthermore, *Pterostylis* species are known to be mainly distributed in Australia [50].

If PtBV does in fact have an Australian origin, then a corollary would be the existence of a native thrips vector. Candidate vectors are *Dichromothrips* spp., a genus of thrips that is widespread on the Orchidaceae in the Old World tropics from Africa to New Zealand [51]. *D. australiae* Mound has been collected from *Pterostylis cycnocephala* Fitzg., *P. nutans* and *P. pedunculata* in the ACT and NSW and *D. spiranthidis* Bagnall has a wider host range within the Orchidaceae including *Microtis unifolia* (G.Forst.) Rchb.f., *Prasophyllum wilkinsoniorum* D.L.Jones, *Spiranthes sinensis* (Pers.) Ames and a *Thelymitra* sp. [52] (L. A. Mound, pers. comm. 18 October 2021). *D. corbetti* Priesner also occurs in Australia as a pest species of cultivated orchids but it is considered non-native, originating in South-East Asia [52]. If in fact a *Dichromothrips* sp. is acting as a vector, it would not facilitate rapid spread, as most adults are flightless (L. A. Mound, pers. comm. 18 October 2021).

PtBV shares a most recent common ancestor with BCaV, another recently described orthotospovirus that was discovered in South Africa [53]. As with PtBV, BCaV does not readily fit into either the Eurasian or American clades of orthotospoviruses, raising the possibility that it might be indigenous to the African continent and has spread from a native plant species into the weedy and introduced plant species, *Barleria cristata* L. Africa and Australia were once connected as parts of the Gondwanan supercontinent. Perhaps PtBV and its closest known relative, BCaV, had a common Gondwanan origin and the current distribution may be explained by continental drift.

PtPV is less likely to be an indigenous virus than PtBV as all poleroviruses except pepper whitefly-borne vein yellows virus are obligatorily transmitted by aphids [54,55], and Australia has a very depauperate native aphid fauna [56]. Furthermore, those aphid species that are native to Australia are very host-specific and none are specialised for feeding on orchids. The polerovirus, turnip yellows virus, which is more typically associated with brassica crops, has been detected in another ground orchid from Victoria, *Diuris pedunculata* R.Br. [18]. However, its known aphid vectors (*Myzus persicae* (Sulzer), *Brevicoryne brassicae* L., *Aphis gossypii* Glover and *Macrosiphum euphorbiae* (Thomas)) are all introduced species [57]. Combining the fact that PtPV has a close evolutionary relationship with PLRV, CYDV-RPV and CYDV-RPS, which are exotic to Australia, it is likely that PtPV is also exotic and transmitted by introduced vectors. Thus, it is predicted that alternative, introduced plant hosts will eventually be identified for PtPV.

As with many previous studies, the research described in this paper has demonstrated the power of high-throughput sequencing technologies to discover new viruses, and to provide comprehensive sequence data to allow their accurate classification into species. PtBV and PtPV may pose threats to economically important plant crops in Australia and it will be important to investigate the possible existence of alternative hosts and to determine whether PtBV can be transmitted by some of the polyphagous pest thrips species such as *Frankliniella occidentalis* (Pergande) or *Thrips tabaci* Lindeman. If an exotic thrips species does turn out to be a more efficient vector of the virus than its proposed original native vector, then new hosts of PtBV may arise and the infection pressure on the orchids could reach a level that poses an existential threat.

## Figures and Tables

**Figure 1 viruses-14-00365-f001:**
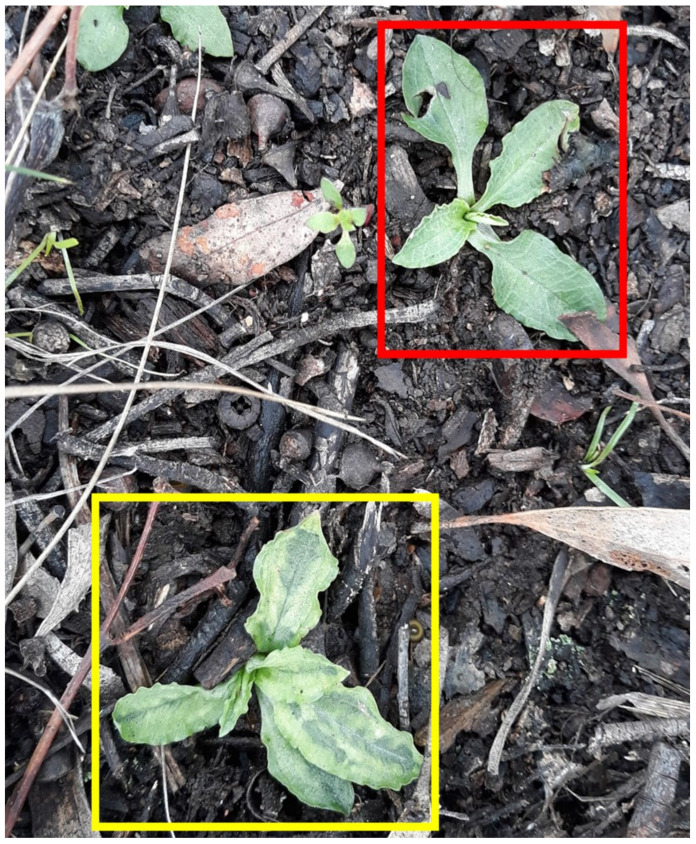
*Pterostylis nutans* infected with Pterostylis blotch virus (yellow frame) and healthy *P. nutans* (red frame), on Black Mountain, the Australian Capital Territory, in 2020.

**Figure 2 viruses-14-00365-f002:**
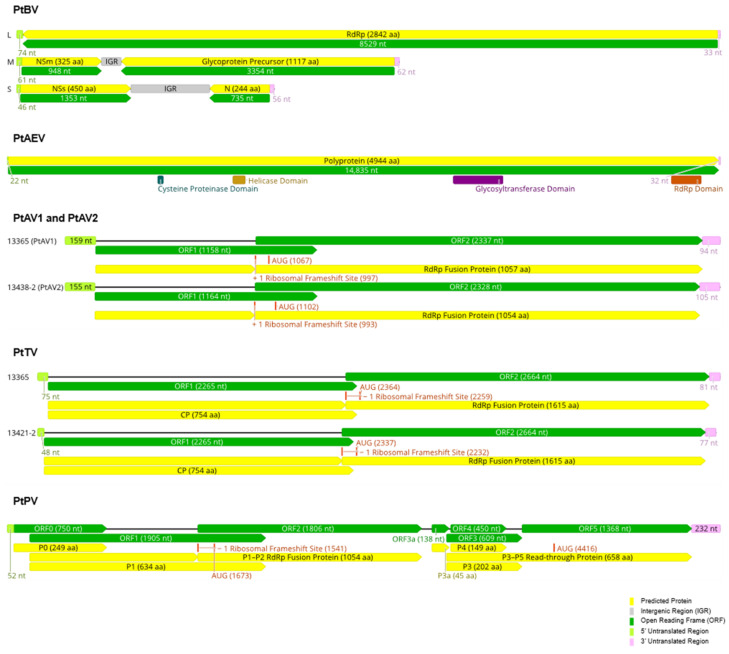
Genome architectures of the viruses identified in this study. Abbreviations are as follows: PtAEV, Pterostylis alphaendornavirus; PtAV1, Pterostylis amalgavirus 1; PtAV2, Pterostylis amalgavirus 2; PtTV, Pterostylis totivirus; PtPV, Pterostylis polerovirus; RdRp, RNA-dependent RNA polymerase; NSm, non-structural protein, M segment; NSs, non-structural protein, S segment; N, nucleocapsid protein; CP, capsid protein; nt, nucleotide; aa, amino acid.

**Figure 3 viruses-14-00365-f003:**
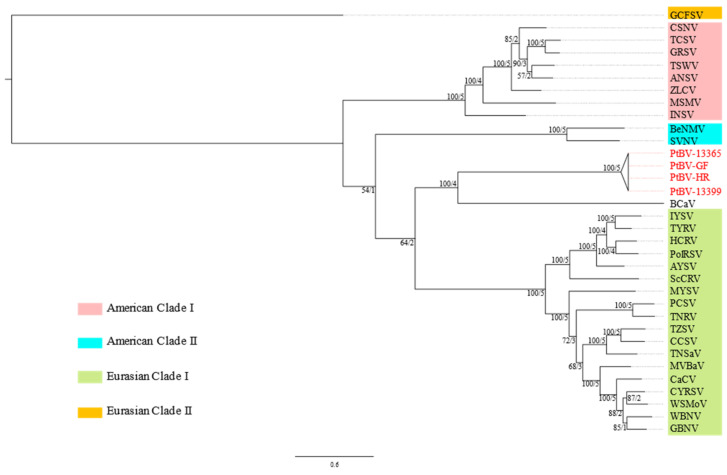
Maximum likelihood reconstruction of the phylogeny of orthotospoviruses based on concatenated sequence alignments of five genes (RNA-dependent RNA polymerase, non-structural protein M segment, glycoprotein precursor, non-structural protein S segment and nucleocapsid protein). The acronyms in red font correspond to the viruses discovered in this study (Table 1) and the other virus acronyms are defined in Appendix A. The numbers to the left of the slash next to the branches are non-parametric bootstrap values based on 1000 replicates; the numbers to the right of the slash are the number of decisive gene trees (5 at maximum) supporting the branch. The nucleotide substitution models used are listed in Appendix A.

**Figure 4 viruses-14-00365-f004:**
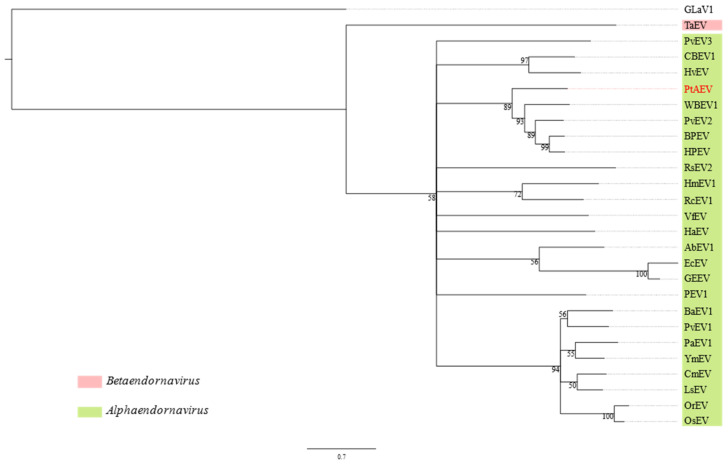
Maximum likelihood reconstruction of the phylogeny of alphaendornaviruses based on the RNA-dependent RNA polymerase gene. The acronym in red font corresponds to the virus discovered in this study (Table 1) and the other virus acronyms are defined in Appendix A. The numbers next to the branches are non-parametric bootstrap values based on 1000 replicates. The nucleotide substitution model used for the phylogenetic tree is listed in Appendix A.

**Figure 5 viruses-14-00365-f005:**
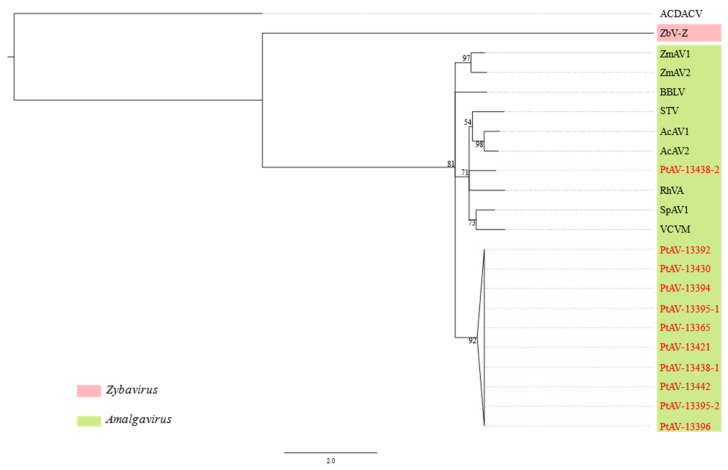
Maximum likelihood reconstruction of the phylogeny of amalgaviruses based on the RNA dependent RNA polymerase gene. The acronyms in red font correspond to the viruses discovered in this study (Table 1) and the other virus acronyms are defined in Appendix A. The numbers next to the branches are non-parametric bootstrap values based on 1000 replicates. The nucleotide substitution model used for the phylogenetic tree is listed in Appendix A.

**Figure 6 viruses-14-00365-f006:**
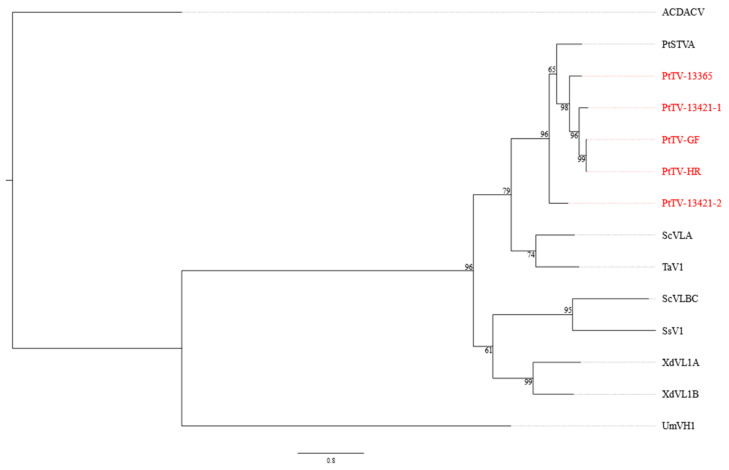
Maximum likelihood reconstruction of the phylogeny of totiviruses based on the RNA-dependent RNA polymerase gene. The acronyms in red font correspond to the viruses s discovered in this study (Table 1) and the other virus acronyms are defined in Appendix A. The numbers next to the branches are non-parametric bootstrap values based on 1000 replicates. The nucleotide substitution model used for the phylogenetic tree is listed in Appendix A.

**Figure 7 viruses-14-00365-f007:**
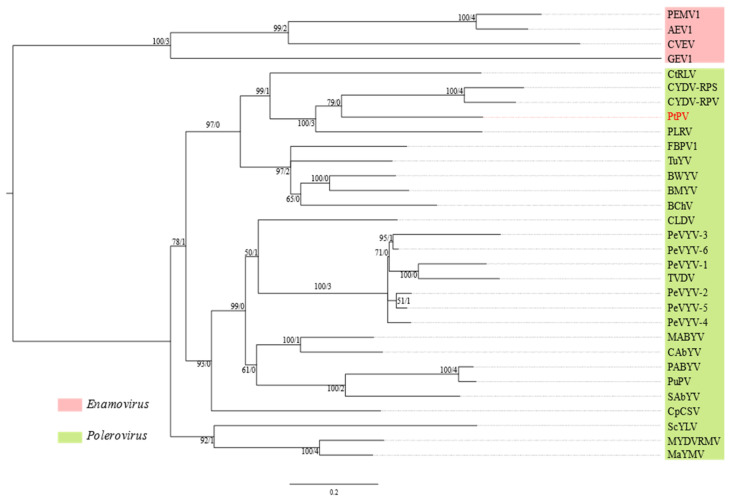
Maximum likelihood reconstruction of the phylogeny of poleroviruses based on concatenated codon sequence alignments of the conserved regions of four genes (serine proteinase, RNA-dependent RNA polymerase, capsid protein (P3) and the domain at the C-terminus of P3 in P3–P5 readthrough protein). The acronym in red font corresponds to the virus isolate discovered in this study (Table 1) and the other virus acronyms are defined in Appendix A. The numbers to the left of the slash next to the branches are non-parametric bootstrap values based on 1000 replicates; the numbers to the right of the slash are the number of decisive gene trees (4 at maximum) supporting the branch. The nucleotide substitution models are listed in Appendix A.

**Table 1 viruses-14-00365-t001:** Orchid Samples Analysed and Types of Viruses Found.

Sample	Location	Host Species	Virus Genus	Sequence Notation	Sequence Depth *
13365	Black Mountain Lookout, ACT	*Pterostylis nutans*	*Amalgavirus*	PtAV-13365	77
			*Orthotospovirus*	PtBV-13365	L: 80,318; M: 54,274; S: 47,192
			*Polerovirus*	PtPV	1,790,464
			*Totivirus*	PtTV-13365	528
13392	Camp Pincham track, Warrumbungle National Park, NSW	*Pterostylis curta*	*Amalgavirus*	PtAV-13392	262
13394	Camp Pincham track, Warrumbungle National Park, NSW	*Pterostylis curta*	*Amalgavirus*	PtAV-13394	198
13395	Camp Pincham track, Warrumbungle National Park, NSW	*Pterostylis curta*	*Amalgavirus*	PtAV-13395-1	95
			*Amalgavirus*	PtAV-13395-2	24
13396	Camp Pincham track, Warrumbungle National Park, NSW	*Pterostylis curta*	*Amalgavirus*	PtAV-13396	53
13399	Burbie Camp track, Warrumbungle National Park, NSW	*Pterostylis curta*	*Orthotospovirus*	PtBV-13399	L:56,527; M: 87,708; S: 41,563
13402	Timor Rock, Warrumbungle National Park, NSW	*Pterostylis curta*	-	-	-
13421	Governor Track, Mt Kaputar National Park, NSW	*Pterostylis nutans*	*Amalgavirus*	PtAV-13421	15
			*Totivirus*	PtTV-13421-1	665
			*Totivirus*	PtTV-13421-2	35
13430	Wellington, Mt Arthur Reserve, NSW	*Pterostylis curta*	*Amalgavirus*	PtAV-13430	112
13438	Grill Cave, Bungonia National Park, NSW	*Pterostylis nutans*	*Amalgavirus*	PtAV-13438-1	40
			*Amalgavirus*	PtAV-13438-2	15
13442	Green track, Bungonia National Park, NSW	*Pterostylis curta*	*Amalgavirus*	PtAV-13442	97
GF	Gibraltar Falls, Namadgi National Park, ACT	*Pterostylis curta*	*Orthotospovirus*	PtBV-GF	L: 12,876; M: 10,559; S: 3453
			*Totivirus*	PtTV-GF	63
HR	Hanging Rock, Tidbinbilla Nature Reserve, ACT	*Pterostylis nutans*	*Alphaendornavirus*	PtAEV	68
			*Orthotospovirus*	PtBV-HR	L: 4026; M: 7010; S: 1195
			*Totivirus*	PtTV-HR	36
PI-1	Pine Island Reserve, ACT	*Oligochaetochilus hamatus*	-	-	-
PI-2	Pine Island Reserve, ACT	*Pterostylis nutans*	-	-	-

* Average sequence depth excluding 5’ and 3’ untranslated regions.

## Data Availability

All sequence data generated in this study have been lodged in GenBank.

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
