# Peer review of "Viruses Infecting Greenhood Orchids (Pterostylidinae) in Eastern Australia"

_viruses, 2022, doi:10.3390/v14020365_

Round 1

Reviewer 1 Report

Dear colleagues,

I read with interest all the corrections for this interesting article, improved, and the comments of the authors, however I still maintain two points of my first review.

I deeply hope to see this good article published soon,

  • Concerning the Authors’ response to my review point 4 (major points; “We believe that putting the raw data in the Results section into the context of the virus groups allows us to focus on the larger picture findings in Discussion”), I still maintain that results should be only “raw” description of the data, some comparisons-discussions-conclusions (assignment of virus “as novel species”, “distinct lineages”…) are actually belonging to the discussion part. The present discussion should be divided in two parts with two sub-titles (one with the several present results parts to be moved in one discussion part, one with the present discussion part), this will make a clear structure to the paper which is missing in the present form.

  • Although I understand that it is an added research to illustrate my remark 3 major point (however sections of leaves are not such a long work to do and do not need a whole pathobiology approach…), it seems to be important to include in the paper the authors’ response (or in shorter terms) to point 2 of the other reviewer that I read in the zip file (“we failed to clarify…”) as more or less linked to my own remark; apparently I had difficulties to find this interesting answer(s) in the new text proposed.

Author Response

We appreciate the reviewer’s feedback. We agree that, in general, interpretation of data should be placed in Discussion. However, in the case of virus species assignment and lineage differentiation, as they are concluded based on certain thresholds/criteria (although not necessarily explicit ones), we suppose that they may be considered like stating statistical significance, which is always done in Results. While we acknowledge that there may be a bit more interpretative elements in our Results especially when touching the virus species demarcation criteria, we argue that the interruption of the current text flow as a result of forcing these elements into Discussion has more cons than pros.

Regarding the analysis of leaf sections, we recognise its importance in understanding aspects of virus replication and cytopathology, but also note this would require a separate and much larger study, which is beyond the scope of this paper. It also requires fresh leaf materials, which are no longer available to us at this moment because of the summer dormancy of the plant. However, we will consider the reviewer’s suggestion in future studies.

We chose not to mention the symptoms of the plants not infected by PtBV because they were not characteristic. Also, we consider it a natural assumption that, if we chose to sample a plant, it must look symptomatic to an extent. However, the confusion of natural senescence and disease symptoms is common, which also demonstrates the importance of technologies like high-throughput sequencing in virus discovery. Establishing the causal links between viruses and plant symptoms, on the other hand, requires a different set of experiments, and thus we only pointed out the “correlation” between PtBV infection and the distinctive symptoms in those infected plants.

Reviewer 2 Report

I believe that the revised manuscript by Chao et al., along with their answers to the comments, clarified the issues that I raised regarding their original paper. The text was easy to follow, interesting and presented all the necessary data.

I have no further comments apart from one, minor, suggestion regarding the ORFs annotation. In my opinion the annotation should start from the FS or the readthrough given that there is no actual AUG start codon for that protein.

Also, the authors could use FSFinder or a similar software in their future work to find frameshifts in sequences.

Author Response

We appreciate the feedback and suggestions from the reviewer. We have reconsidered the reviewer’s comment on the annotation of ORF and accordingly made some changes to Figure 2. We now have annotated the starting point of the second ORF in a fusion protein at the site of frameshift/read-through and made a separate mark for the AUG position so that the ORF annotation better represents the biological properties.

This manuscript is a resubmission of an earlier submission. The following is a list of the peer review reports and author responses from that submission.

Round 1

Reviewer 1 Report

The manuscript by Chao et al., titled “Surveys for Viruses Infecting Greenhood Orchids (Pterostylidinae) in Eastern Australia” describes the identification and initial characterization of six new viruses present in orchids in Australia. The authors collected 15 samples from 11 different locations and used HTS to find the viruses present in each sample. Afterwards they confirmed the 5’ and 3’ genome ends of a selected number of isolates and performed a phylogenetic analysis to illustrate the distinct phylogenetic groupings of these viruses which support their classification as novel species, in accordance to ICTV’s species demarcation criteria.

The paper is well written and easy to follow, however it feels compressed in terms of the data presented in the text, and personally I feel that it lies somewhere between concatenated first reports, a survey, and a diversity analysis without actually completing any of these tasks.

In my opinion the major drawbacks of this manuscript are the lack of a clear scope, the amount of data presented and the fact that the authors didn’t confirm the presence of the viruses found with HTS in each respective sample. More specifically:

As stated in the abstract the main aim was to characterize PtBV. However, the virus was identified in 4 samples and the authors didn’t provide any data about the similarity of these isolates between them, except that they are 99% identical in the N gene (line 283). No actual characterization was performed apart from the fact that PtBV is a new species in the genus Orthotospovirus. It would be really interesting to know if these isolates exhibit any genetic diversity given that they are from two different plant species and from different locations which are considerably far apart. The rest of the manuscript describes the rest of the viruses found, in a similar way. Moreover, the authors don’t provide any data about the symptoms present in each sample. It is only mentioned that one plant was symptomatic (sample 13365) (line 227) however PtBV was found in 4 samples. Based on what criteria were these samples collected? Randomly? I believe that more information and focus should be given in this aspect so an initial correlation with the disease could be given according to the viruses present in each sample. As it is, if 13365 was the only symptomatic sample it is probable that the polerovirus is the causal agent.

Regarding the amount of data, I believe that the authors should present identity percentages for each new virus at least with the closest species and the presence of genomic traits if they exist. Phylogenetic trees are useful for groupings and classification, but more data should be used for the final classification. Of course, the authors do mention the demarcation criteria these new virus fulfill, however I believe that for orthotospoviruses and poleroviruses which could reassort and recombine, more data is necessary.

The fact that the authors didn’t confirm the presence of the viruses in each sample is troublesome for me. The 5’ and 3’ RACE could be considered a confirmation but even those weren’t performed for every isolate and in some cases the authors couldn’t acquire the amplicons (lines 336, 343-345) which is also a problem. I believe that the presence of every virus should be confirmed by RT-PCR and Sanger sequencing, which in turn could also confirm the sequence which was obtained from the de novo assembly.

Some other points that the authors should correct/reconsider are:

In figure 2, the position of ORFs. When a frameshift or readthrough is present, the ORF continues from this point and not from the next ATG codon. This should be corrected in PtAV1/2, PtTV and PtPV and in some parts in the text (line 346). Moreover, the method the authors used to identify the frameshift site should be included in the text (eg. line 319).

Figure 3 is too small/poor quality, to actually see the numbers and names. Please adjust it accordingly. Moreover, BeNMV and SVNV are considered a new world (American) clade and GCFSV as an Old World (Asian) one (Oliver and Whitfield, 2016).

The rationale of the phylogenetic reconstruction should be explained. In my opinion using concatenated conserved regions for viruses that are known to recombine is wrong, because each ORF could have a completely different origin, and this could lead to a wrong phylogenetic hypothesis. For example, the trees in the supplementary file have different topologies for both PtBV (Figure S2, C and D) and PtPV (Figure S3, C and D). However, I am not familiar with the method used in this paper so it would be helpful if the authors could discuss their rational for manually excluding regions of the genome, the concatenation of genomic regions and the genealogical concordance.

The discussion relies heavily on hypotheses not supported by the data present in the text. For example, in my opinion the presence of a long branch is not an indication of an indigenous virus (lines 430-433). Moreover, PtBV is unlikely to have speciated 550-30 Ma when Gondwanan was formed and separated, and a different set of experiments are necessary to claim this. Similarly, PtPV is said not to be an indigenous virus based on the fact that Australia has a depauperate native aphid fauna, however the fact that the plant was infected means that there is a vector, which is further supported by the fact that another polerovirus was found in orchids in Australia (Wylie et al., 2013). A more extensive survey/study is needed to acquire more biological data which could indicate the origin of PtPV.

A few minor comments are given bellow:

Lines 139-140: Please indicate the software you used to trim the reads. Was the final length chosen arbitrarily?

Lines 210-211: Were PI-1 and 2 symptomatic? Please clarify.

Line 232: Number is missing after the citation (15).

Lines 396-399: The fact that PtPV clusters consistently with CYDV but the P3 is more similar to PLRV is contradictory and could be an indication of recombination and of a problem with the phylogenetic trees. Please correct/rephrase this part.

Overall, the paper is very interesting considering the impact these viruses could have in native flora, even more so if they are introduced from outside Australia. However, I strongly believe that the authors should present the data in a better way, which could strengthen their results.

Reviewer 2 Report

Dear colleagues,

This review is concerning a research work entitled “Surveys for Viruses Infecting Greenhood Orchids (Pterostylidinae) in Eastern Australia, by Hsu-Yao Chao, Mark A. Clements, Anne Mackenzie, Ralf G. Dietzgen, John E. Thomas and Andrew D. W. Geering. As detailed experiments, I recommend it for an international audience in this journal, however several points have to be precised and a major revision is requested.

Please notice that the five major points of my comments (at the beginning) are very important (mandatory…) for a suitable value of the article, in order to bring a broader audience to this article and to this journal for specialists and non-specialists. Minor points are also enhanced at the end of this review.

I deeply hope to see this good article published soon,

The five major point are:

  • 1- Concerning points 3.2, 3.3, 3.4, as these viruses seem to be well defined, it is not extremely clear why these viruses appear as “novel” with apparently no official recognition (taxonomical rank…) by the authors (on the other hand, PtBV is said “previously reported but uncharacterized virus”); this should be much more explained in the discussion to clarify this point, at least for non-specialists;

  • 2- As a botanist I am, I am very sensible to latin names; in order to provide a homogeneous text all plants cited from the species level (and below)  should be followed by their author(s) abbreviations, at least the first time each latin name appears in the text; in the present form of the text it is too heterogeneous (consult International Plant Names Index https://www.ipni.org); concerning viruses names I am not extremely familiar with some names but as in other papers some variations seem to exist for the same taxon, an international list should be also consulted (ictv https://talk.ictvonline.org ??);

  • 3- Although this paper is focused on the virus genome, concerning the disease caused (?) by the virus, necrosis is a too general symptom evocated with almost no precise meaning; moreover the authors are interestingly very cautious as several (putative) viruses seem to be involved; it should be welcome to provide at least some sections (it is very easy to do) showing the precise damages in epidermis cells, xylem, phloem, palisade parenchyme…;

  • 4- Results should be only “raw” description of the data, all comparisons belonging to the discussion part; in this respect, the present results (usually the first paragraph of each sub-point of part 3) contain many discussion parts (the other paragraphs of each sub-point), which should be moved in the discussion part;

  • 5- References already taken in account by the authors are of great interest, however checking briefly in the word of science WOS with key-words and topics (of course largely selected here as topics sensu lato) like the latin or english names of the taxa and the technique (s) involved, other articles appear and references should be apparently once more selected (if relevant…) and used in order to provide a larger view of this interesting research. Among these are the followings:

 [1-8]

  1. Cheng, H.W.; Tsai, W.T.; Hsieh, Y.Y.; Chen, K.C.; Yeh, S.D. Identification of a Common Epitope in Nucleocapsid Proteins of Euro-America Orthotospoviruses and Its Application for Tagging Proteins. Int J Mol Sci 2021, 22, doi:ARTN 858310.3390/ijms22168583.

  1. Dawson, M.I.; Molloy, B.P.J.; Beuzenberg, E.J. Contributions to a chromosome atlas of the New Zealand flora-39. Orchidaceae. New Zeal J Bot 2007, 45, 611-684, doi:Doi 10.1080/00288250709509743.

  1. Fuji, S.; Inoue, M.; Yamamoto, H.; Furuya, H.; Naito, H.; Matsumoto, T. Nucleotide sequences of the coat protein gene of potyviruses infecting Ornithogalum thyrsoides. Archives of Virology 2003, 148, 613-621, doi:10.1007/s00705-002-0961-9.

  1. Gautam, S.; Mugerwa, H.; Sundaraj, S.; Gadhave, K.R.; Murphy, J.F.; Dutta, B.; Srinivasan, R. Specific and Spillover Effects on Vectors Following Infection of Two RNA Viruses in Pepper Plants. Insects 2020, 11, doi:ARTN 60210.3390/insects11090602.

  1. Janes, J.K.; Duretto, M.F. A new classification for subtribe Pterostylidinae (Orchidaceae), reaffirming Pterostylis in the broad sense. Aust Syst Bot 2010, 23, 260-269, doi:10.1071/Sb09052.

  1. Janes, J.K.; Steane, D.A.; Vaillancourt, R.E.; Duretto, M.F. A molecular phylogeny of the subtribe Pterostylidinae (Orchidaceae): resolving the taxonomic confusion. Aust Syst Bot 2010, 23, 248-259, doi:10.1071/Sb10006.

  1. Ong, J.W.L.; Li, H.; Sivasithamparam, K.; Dixon, K.W.; Jones, M.G.K.; Wylie, S.J. The challenges of using high-throughput sequencing to track multiple bipartite mycoviruses of wild orchid-fungus partnerships over consecutive years. Virology 2017, 510, 297-304, doi:10.1016/j.virol.2017.07.031.

  1. Wu, X.J.; Xu, S.; Zhao, P.Z.; Zhang, X.; Yao, X.M.; Sun, Y.W.; Fang, R.X.; Ye, J. The Orthotospovirus nonstructural protein NSs suppresses plant MYC-regulated jasmonate signaling leading to enhanced vector attraction and performance. Plos Pathog 2019, 15, doi:ARTN e100789710.1371/journal.ppat.1007897.

As minor points:

1 in the title, delete "surveys for"? (a short title is usually more attractive for readers);

2 in point 2.1, put full latin names of the taxa as it is another chapter;

3 the phylogenetic trees of figure 3 are too small, the letters-abbreviations on the right side are almost not visible;

4 in the discussion part, precise some values of the phylogenetic trees (branches and etc, as provided in Fig. 3), it will sustain much more objectively the interesting remarks;

5 apparently the references are not in the proper format for this journal, please consult recommendations for this journal and check in the articles recently published.